# Elliptic Flow and Its Fluctuations from Transport Models for Au+Au Collisions at $\sqrt{s_{NN}}$ = 7.7 and 11.5 GeV

**Vinh Ba Luong** [1,2,3,*] (ID), **Dim Idrisov** [2] (ID), **Petr Parfenov** [2,*] (ID) **and Arkadiy Taranenko** [2] (ID)

1  Veksler and Baldin Laboratory of High Energy Physics, Joint Institute for Nuclear Research, 6 Joliot-Curie, Dubna 141980, Russia
2  Physics of Condensed Matter Department, National Research Nuclear University MEPhI, 31 Kashirskoe Highway, Moscow 115409, Russia
3  Dalat Nuclear Research Institute, Vietnam Atomic Energy Institute, 1 Nguyen Tu Luc, Dalat 670000, Vietnam
*  Correspondence: vinhbluong@jinr.ru (V.B.L.); peparfenov@mephi.ru (P.P.)

**Abstract:** The elliptic flow $v_2$ is one of the key observables sensitive to the transport properties of the strongly interacting matter formed in relativistic heavy-ion collisions. In this work, we report on the calculations of $v_2$ and its fluctuations of charged hadrons produced in Au+Au collisions at center-of-mass energy per nucleon pair $\sqrt{s_{NN}}$ = 7.7 and 11.5 GeV from several transport models and provide a direct comparison with published results from the STAR experiment. This study motivates further experimental investigations of $v_2$ and its fluctuations with the Multi-Purpose Detector (MPD) at the NICA Collider.

**Keywords:** heavy-ion collisions; elliptic flow; fluctuations; MPD experiment; NICA

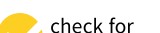



## 1. Introduction

In relativistic heavy-ion collisions, momentum distributions of the produced particles are anisotropic in the transverse plane perpendicular to the beam direction, well-known as the anisotropic flow. The elliptic flow, $v_2 = \langle \cos[2(\phi - \Psi_n)] \rangle$, is the second harmonic coefficient in the Fourier expansion of the azimuthal distribution of particle transverse momentum ($dN/d(\phi - \Psi_n)$) [1,2], where $\phi$ is the azimuthal angle of the particle of a given type and $\Psi_n$ is the azimuthal angle of the event plane. The elliptic flow $v_2$ signal of the produced particles carries important information on the pressure gradients, the equation of state (EOS), the transport coefficients of the medium, and initial conditions in heavy-ion collisions. It has been extensively studied both at the Relativistic Heavy Ion Collider (RHIC) and Large Hadron Collider (LHC) energies. The observed large $v_2$ for all produced particles suggested that the system formed in heavy-ion collisions at RHIC and LHC is a strongly coupled Quark Gluon Plasma (sQGP) [3]. For low transverse momentum ($p_T < 2$–3 GeV/$c$), $v_2(p_T)$ of the produced particles is well described by viscous hydrodynamic models and the overall good agreement between the data and model calculations can be reached for small values of specific shear viscosity $\eta/s$ close to the lower conjectured bound of $1/4\pi$ [4]. In this model framework, the values of the $v_2$ coefficient results from the evolution of the sQGP, driven by the spatial anisotropy of its initial energy density profile, characterized by finite eccentricity moment $\varepsilon_2$. The elliptic flow correlates almost linearly with the initial eccentricity $\varepsilon_2$: $v_2 = \kappa_2\varepsilon_2$, where the proportionality coefficient $\kappa_2$ encodes the medium response, which is sensitive to $\eta/s$. The nucleon distribution in the collision overlap area fluctuates event-by-event, which results in the fluctuations of eccentricity $\varepsilon_2$ and, in its turn, the $v_2$ fluctuations via—$\kappa_2$ [5–7]. The precision extraction of $\eta/s$ requires the model constraints for fluctuations of $v_2$ and $\varepsilon_2$. The relative fluctuation of $v_2$ can be quantified by the ratio of the first two multi-particle cumulants [8,9], namely the $v_2\{4\}/v_2\{2\}$ ratio. In particular, the larger the fluctuations of $v_2$ are, the smaller the ratio $v_2\{4\}/v_2\{2\}$ is. In addition to the consideration of fluctuations, flow measurements require the suppression

of the so-called non-flow correlations [10]. They include the overall transverse momentum conservation, small angle azimuthal correlations due to final state interactions, resonance decays, mini-jet production, and quantum correlations due to the HBT effect. The methods of multi-particle cumulants and Lee–Yang zeros [11,12] have been developed to suppress the non-flow correlations.

In this work, we report on the calculations of the elliptic flow and its fluctuations of charged hadrons produced in Au+Au collisions at $\sqrt{s_{NN}}$ = 7.7 and 11.5 GeV. The calculations have been carried out using the current state-of-art Monte Carlo models of heavy-ion collisions: UrQMD, SMASH, AMPT, and the hybrid vHLLE+UrQMD. The selected collision energies allow one to provide a direct comparison of the obtained results with the published $v_2$ values from the Beam Energy Scan (BES) program of the STAR experiment at RHIC [13,14]. The MPD experiment [15,16] at the NICA collider [17] is planned to start studying the heavy-ion collisions at $\sqrt{s_{NN}}$ = 4–11 GeV in 2023. The primary scientific mission of the MPD experiment is to investigate the properties of strongly interacting matter at high net-baryon densities. Different methods for elliptic flow measurements have been used to investigate the anticipated physics performance of the MPD detector system and the contribution of non-flow correlations and flow fluctuations at NICA energies.

## 2. Monte Carlo Models

In this work, we have used several Monte Carlo models to simulate Au+Au collisions at energies: $\sqrt{s_{NN}}$ = 7.7 and 11.5 GeV. Brief descriptions of these models are given below:

The Ultrarelativistic Quantum Molecular Dynamics (UrQMD) model [18,19] is a microscopic transport approach which describes hadronic reactions at low and intermediate energies in terms of collisions among hadrons and their resonances. At incident energies above $\sqrt{s_{NN}}$ = 5 GeV, the multiparticle production is dominated by the excitation of color strings and their subsequent fragmentation into hadrons. We have used the cascade mode of UrQMD (version 3.4). The previous works dedicated to the elliptic flow at a wide energy range applied the UrQMD model either with or without the relativistic hydrodynamic description [20,21]. The $v_2$ fluctuations in the UrQMD model were also investigated at top RHIC energy [22].

The Simulating Many Accelerated Strongly-interacting Hadrons (SMASH) [23] is a relativistic hadronic transport approach including all well-established hadrons up to a mass of ∼2 GeV as degrees of freedom. Most interactions proceed via resonance excitation and decay at lower energies or string excitation and fragmentation at higher energies. Free parameters of the string excitation and decay are tuned to match the experimental measurements in inelastic p+p collisions. We have used version 1.7 of the model with a default set of parameters.

The string melting version of A Multi-Phase Transport model (AMPT-SM) [24] uses the heavy ion jet interaction generator (HIJING) for the initial conditions, Zhang's parton cascade (ZPC) for modeling partonic scatterings and the quark coalescence model for hadronization. After hadronization, the hadronic interactions are modelled by the ART (A Relativistic Transport) model, which incorporates both elastic and inelastic scattering for baryon–baryon, baryon–meson, and meson–meson interactions. We have generated events with the AMPT-SM (version ampt-v1.26t7-v2.26t7) model for two values of partonic cross section: $\sigma_p$ = 1.5 and 0.8 mb.

vHLLE+UrQMD is a viscous hybrid model employing the UrQMD hadron/string cascade transport model for the early and late non-equilibrium stages of the reaction, and (3+1)-D dimensional relativistic viscous hydrodynamic code vHLLE [25,26] for the quark-gluon plasma phase. The equation of state based on the Chiral model in the fluid stage (XPT EOS) has been used. It has a crossover type transition between QGP and hadronic phases for all baryon densities. Fluid to particle transition, or particlization, takes place when the energy density $\epsilon$ in the hydro cells reaches the switching value $\epsilon_{SW}$ = 0.5 GeV/$fm^3$. The hadronic rescatterings and decays are treated with the UrQMD hadronic cascade. The initial state parameters, hydrodynamic starting time $\tau_0$ and specific shear viscosity $\eta/s$ in fluid

phase are tuned for different collision energies in order to reproduce basic experimental bulk observables in the RHIC Beam Energy Scan: (pseudo) rapidity distributions, transverse momentum spectra and elliptic flow coefficient for inclusive charged hadrons. For more details see Table II in Ref. [26].

The UrQMD and SMASH models only take the hadronic interactions into consideration, while the AMPT SM and vHLLE+UrQMD models incorporate both partonic and hadronic interactions.

In total, a sample of ~60 M minimum bias Au+Au events has been generated by each model at $\sqrt{s_{NN}}$ = 7.7 and 11.5 GeV, respectively. The generated events were used to analyse the elliptic flow signals of the charged hadrons by way of different methods.

In order to be consistent with the analysis of the STAR data, the centrality definition is based on the measured charged particle multiplicity from the TPC within pseudo-rapidity $|\eta| < 0.5$, uncorrected for detection efficiencies. The measured multiplicity distribution has been compared with a Monte-Carlo Glauber Model [27] to extract the centrality of each event as in our previous work [28]. The collision centralities of events are classified according to fractions of the total inelastic cross section. The 0–10% centrality interval corresponds to the most central collisions (i.e., events with a small impact parameter), while the 70–80% interval represents peripheral collisions (i.e., events with a large impact parameter).

## 3. Methods of Elliptic Flow Analysis

Several methods have been used in order to measure the elliptic flow coefficients of charged hadrons in Au+Au collisions at $\sqrt{s_{NN}}$ = 7.7 and 11.5 GeV: the TPC event plane ($v_2\{\Psi_{2,\text{TPC}}\}$), the TPC scalar product ($v_2^{\text{SP}}\{Q_{2,\text{TPC}}\}$), the FHCal event plane ($v_2\{\Psi_{1,\text{FHCal}}\}$), the FHCal scalar product ($v_2^{\text{SP}}\{Q_{1,\text{FHCal}}\}$), the Lee–Yang zeros ($v_2\{\text{LYZ}\}$), and the two-, four-, six-particle cumulant ($v_2\{2\}$, $v_2\{4\}$, $v_2\{6\}$) methods.

The Lee–Yang zeros method [11,12] considers correlations of all particles in the event. Therefore, it is expected to produce the cleanest values of the genuine anisotropic flow. The product generating function of the Lee–Yang zeros method is defined as

$$G^\theta(\mathrm{i}r) = \left\langle \prod_{j=1}^{M}\left[1 + \mathrm{i}r\omega_j \cos(2(\phi_j - \theta))\right]\right\rangle, \tag{1}$$

where $r$ is a positive real variable, $\theta$ is an arbitrary azimuthal projection angle, and $\omega_j$ is the particle weight. We used the $1/M$ weight to reduce the effect of multiplicity fluctuations. Five $\theta$ values were taken to reduce the statistical uncertainties. We checked that increasing the number of $\theta$ above five did not provide an improvement of statistical fluctuations. For the demonstration, Figure 1 shows the squared modulus of the generating function $|G^\theta(\mathrm{i}r)|^2$ as a function of $r$ for the case of $\theta = \pi/5$ in four different centrality classes.

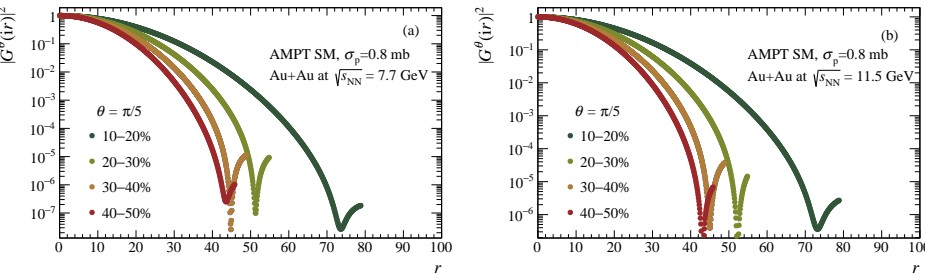

**Figure 1.** Squared modulus of the second harmonic product generating function $|G^\theta(ir)|^2$ of Lee–Yang zeros method as a function of $r$ for $\theta = \pi/5$ for four centrality classes from Au+Au collisions at $\sqrt{s_{NN}}$ = 7.7 (**a**) and 11.5 GeV (**b**) using AMPT SM model.

After finding the position $r_0^\theta$ of the first zero of $|G^\theta(ir)|^2$, $v_2$, for the case of $1/M$ weights, is determined by the Lee–Yang zeros method as follows:

$$v_2\{\text{LYZ}\} = \left\langle \frac{j_{01}}{r_0^\theta} \right\rangle, \tag{2}$$

where $j_{01} = 2.405$ is the first root of the Bessel function $J_0$ and the $\langle \dots \rangle$ brackets denote the average taken over five $\theta$ angles. Equations for differential $v_2\{\text{LYZ}\}(p_T)$ can be found in Refs. [11,12]. The Lee–Yang zeros method is sensitive to multiplicity fluctuations and it only works for a sufficiently large signal-to-noise ratio. Since the signal is $v_2$ and the noise is proportional to $1/\sqrt{M}$, the parameter $\chi = v_2\sqrt{M}$ determines the applicability of the method. In the AMPT SM model, $\chi > 0.8$ only for 10–40% centrality classes in Au+Au collisions at $\sqrt{s_{NN}} = 7.7$ and 11.5 GeV. It was found that $\chi < 0.8$ leads to a large scattering of the results [10]. Despite the advantages of non-flow suppression, the Lee–Yang zeros method fails to measure the flow for central collisions due to a small $v_2$ signal and peripheral collisions—due to the small $M$ values.

For the other flow measurement methods, the unit and the flow vectors are defined as

$$u_{n,j} = e^{in\phi_j} = (\cos n\phi_j, \sin n\phi_j), \ Q_n = \sum_{j=1}^{M} w_j u_{n,j}, \tag{3}$$

where $\phi_j$ is the azimuthal angle in the momentum space of the particles, $M$ denotes multiplicity of the particles included in the given flow vector and $w_j$ is the $j$th-particle weight.

In the $Q$-cumulants method [29], the two-, four- and six-particle correlations previously proposed in Refs. [8,9] can be expressed in terms of flow vectors $Q_n$ as follows:

$$\langle 2 \rangle = \left\langle e^{in(\phi_1 - \phi_2)} \right\rangle = \frac{|Q_n|^2 - M}{M(M-1)},$$

$$\langle 4 \rangle = \left\langle e^{in(\phi_1 + \phi_2 - \phi_3 - \phi_4)} \right\rangle$$
$$= \{|Q_n|^4 + |Q_{2n}|^2 - 2Re(Q_{2n}Q_n^*Q_n^*) - 2[2(M-2)|Q_n|^2$$
$$- M(M-3)]\}/[M(M-1)(M-2)(M-3)],$$

$$\langle 6 \rangle = \left\langle e^{in(\phi_1 + \phi_2 + \phi_3 - \phi_4 - \phi_5 - \phi_6)} \right\rangle \tag{4}$$
$$= [|Q_n|^6 + 9|Q_{2n}|^2|Q_n|^2 - 6Re(Q_{2n}Q_nQ_n^*Q_n^*Q_n^*)$$
$$+ 4Re(Q_{3n}Q_n^*Q_n^*Q_n^*) - 12Re(Q_{3n}Q_{2n}^*Q_n^*) + 18(M-4)$$
$$\times Re(Q_{2n}Q_n^*Q_n^*) + 4|Q_{3n}|^2 - 9(M-4)(|Q_n|^4 + |Q_{2n}|^2)$$
$$+ 18(M-2)(M-5)|Q_n|^2 - 6M(M-4)(M-5)]$$
$$/[M(M-1)(M-2)(M-3)(M-4)(M-5)].$$

Then, following the general cumulant formalism, the second-, fourth-, sixth-order cumulants can be given by the following:

$$c_n\{2\} = \langle\langle 2 \rangle\rangle, \ c_n\{4\} = \langle\langle 4 \rangle\rangle - 2\langle\langle 2 \rangle\rangle^2, \ c_n\{6\} = \langle\langle 6 \rangle\rangle - 9\langle\langle 2 \rangle\rangle\langle\langle 4 \rangle\rangle + 12\langle\langle 2 \rangle\rangle^3, \tag{5}$$

where the double brackets denote the weighted average of multi-particle correlations. The weights are the total number of combinations from two-, four-, or six-particle correlations, respectively.

We divided the $\eta$ range of the TPC detector into two sub-events: $A$ ($-1.5 < \eta^A < -0.05$) and $B$ ($0.05 < \eta^B < 1.5$), separated by a pseudo-rapidity gap $|\Delta\eta| > 0.1$ in order to suppress the non-flow from the short range correlations. Then, $\langle 2 \rangle$ in Equation (4) is modified to be:

$$\langle 2 \rangle_{|\Delta\eta|} = \frac{Q_n^A \cdot Q_n^B}{M^A \cdot M^B}, \tag{6}$$

where $Q_n^A$ and $Q_n^B$ are the flow vectors from sub-events $A$ and $B$, and $M^A$ and $M^B$ are the corresponding sub-event multiplicities.

The harmonic flow $v_n$ can be estimated via the cumulants of different orders ($n$ = 2, 3, 4, ...),

$$v_n\{2\} = \sqrt{c_n\{2\}}, \ v_n\{4\} = \sqrt[4]{-c_n\{4\}}, \ v_n\{6\} = \sqrt[6]{c_n\{6\}/4}. \tag{7}$$

Finally, estimations of differential flow can be expressed as follows:

$$v_n'\{2\} = \frac{d_n\{2\}}{\sqrt{c_n\{2\}}}, \ v_n'\{4\} = \frac{d_n\{4\}}{-c_n\{4\}^{3/4}}, \tag{8}$$

where $d_n\{2\}$ and $d_n\{4\}$ are the two- and four-particle differential cumulants as defined in Ref. [29].

The TPC scalar product method [10] correlates the azimuthal angle $\phi_j$ of each particle of interest from TPC sub-event $A$ with the flow vector $Q_2$ of the TPC sub-event $B$ and vice-versa:

$$v_2^{SP}\{Q_{2,\text{TPC}}\} = \frac{\left\langle u_{2,j}^{A(B)} \cdot Q_{2,\text{TPC}}^{B(A),*} \right\rangle}{\sqrt{\left\langle Q_{2,\text{TPC}}^A \cdot Q_{2,\text{TPC}}^{B,*} \right\rangle}}. \tag{9}$$

The pseudo-rapidity gap of $|\Delta\eta| > 0.1$ between two sub-events is used to suppress the non-flow effect from short range correlations.

In the TPC event plane method [2], the flow coefficient can be estimated by using Equation (9), where $Q_n$ is replaced by its unit vector $Q_n/|Q_n|$ as follows:

$$v_2\{\Psi_{2,\text{TPC}}\} = \frac{\left\langle u_{2,j}^{A(B)} \cdot \frac{Q_{2,\text{TPC}}^{B(A),*}}{\left|Q_{2,\text{TPC}}^{B(A)}\right|} \right\rangle}{\sqrt{\left\langle \frac{Q_{2,\text{TPC}}^A}{\left|Q_{2,\text{TPC}}^A\right|} \cdot \frac{Q_{2,\text{TPC}}^{B,*}}{\left|Q_{2,\text{TPC}}^B\right|} \right\rangle}} = \frac{\left\langle \cos\left[2\left(\phi_j^{A(B)} - \Psi_{2,\text{TPC}}^{B(A)}\right)\right] \right\rangle}{R_2(\Psi_{2,\text{TPC}})}, \tag{10}$$

where $\Psi_{2,\text{TPC}} = \tan^{-1}(Q_{2,y}/Q_{2,x})$ is the TPC event plane of the second order harmonic and $R_2(\Psi_{2,\text{TPC}}) = \sqrt{\langle\cos[2(\Psi_{2,\text{TPC}+} - \Psi_{2,\text{TPC}-})]\rangle}$ is the event plane resolution. While it is practically preferable to measure the flow by the event plane method, it has been proven to be ambiguous [7,10]: $v_2\{\Psi_{2,\text{TPC}}\}^2 = \langle v \rangle^2 + (\alpha - 1)\sigma_v^2$, where $\sigma_v^2 = \langle v^2 \rangle - \langle v \rangle^2$ is the magnitude of flow fluctuations and $\alpha$ is a coefficient depending on the event plane resolution ($1 \leq \alpha \leq 2$). $\alpha \to 1$ when the resolution $R_2$ is high, while $\alpha \to 2$ corresponds to low $R_2$, which is expected to be the case in the MPD experiment.

For the methods using FHCal detectors with the pseudo-rapidity coverage of $2 < |\eta| < 5$, the left (right) FHCal flow vector and the full FHCal event plane of the first order harmonic are calculated as follows:

$$Q_1^{L(R)} = \sum_{modules} E_j e^{i\phi_j}, \ \Psi_{1,\text{FHCal}}^{\text{full}} = \tan^{-1}\left(\frac{Q_{1,y}^L + Q_{1,y}^R}{Q_{1,x}^L + Q_{1,x}^R}\right), \tag{11}$$

where $\phi_j$ is the azimuthal angle of the center of the $j$th FHCal module in the transverse plane, and $E_i$ is the energy deposition in the $j$th module of FHCal (weight to improve the event plane resolution and maximize the contribution of spectators to the flow vector). Then, $v_2$ is calculated via the FHCal scalar product and event plane methods by the following equations:

$$v_2^{SP}\{Q_{1,\text{FHCal}}\} = \frac{\left\langle u_{2,j} \cdot Q_1^{L,*} \cdot Q_1^{R,*} \right\rangle}{\left\langle Q_1^L \cdot Q_1^{R,*} \right\rangle}, \ v_2\{\Psi_{1,\text{FHCal}}\} = \frac{\left\langle \cos\left[2\left(\phi_j - \Psi_{1,\text{FHCal}}^{\text{full}}\right)\right] \right\rangle}{R_2(\Psi_{1,\text{FHCal}}^{\text{full}})}, \tag{12}$$

where $u_{2,j}$ and $\phi_j$ are taken from particles within TPC acceptance ($|\eta| < 1.5$) and $R_2(\Psi^{\text{full}}_{1,\text{FHCal}})$ is the event plane resolution of the elliptic flow with respect to the first order harmonic event plane [2].

Following the above-mentioned descriptions, the pseudo-rapidity gaps of $|\Delta\eta| > 0.1$ between the two TPC sub-events for $v_2\{\Psi_{2,\text{TPC}}\}$, $v_2^{\text{SP}}\{Q_{2,\text{TPC}}\}$, $v_2\{2\}$ and $|\Delta\eta| > 0.5$ between the TPC and FHCal detectors for $v_2^{\text{SP}}\{Q_{1,\text{FHCal}}\}$, $v_2\{\Psi_{1,\text{FHCal}}\}$ are applied to suppress non-flow effects. If one neglects residual non-flow contributions, for a Bessel–Gaussian distribution of $v_2$: $P(v) = BG(v; v_0, \sigma)$, the measured elliptic flow signal can be expressed via the following equations [6,10]:

$$v_2\{\Psi_{2,\text{TPC}}\} \approx v_2^{\text{SP}}\{Q_{2,\text{TPC}}\} = v_2\{2\} = \left\langle v^2 \right\rangle^{1/2} = \sqrt{v_0^2 + 2\sigma^2},$$
$$v_2\{4\} = v_2\{6\} = v_2\{\text{LYZ}\} = \langle v \rangle = v_0 = \left\langle v_2^{\text{RP}} \right\rangle. \tag{13}$$

Thus, one can exploit the $v_2\{4\}/v_2\{2\}$ ratio to investigate the $v_2$ fluctuations [30], A large contribution from $v_2$ fluctuations will result in $v_2\{4\}/v_2\{2\} < 1$, while the weak one gives $v_2\{4\}/v_2\{2\} \sim 1$.

Alternatively, for strong correlations between the spectator plane and the reaction plane, one can expect the following relations:

$$v_2\{\Psi_{1,\text{FHCal}}\} = v_2^{\text{SP}}\{Q_{1,\text{FHCal}}\} = \left\langle v_2^{\text{RP}} \right\rangle. \tag{14}$$

## 4. Results and Discussion

In Figure 2, the $v_2\{\Psi_{2,\text{TPC}}\}$, $v_2\{2\}$, $v_2\{4\}$ of charged hadrons measured in the STAR experiment are compared with the ones obtained from the models. At both $\sqrt{s_{NN}} = 7.7$ and 11.5 GeV, the UrQMD and SMASH models could not reproduce the observed $v_2$ signal. In contrast, the vHLLE+UrQMD and AMPT SM models give a reasonable agreement with the experimental data.

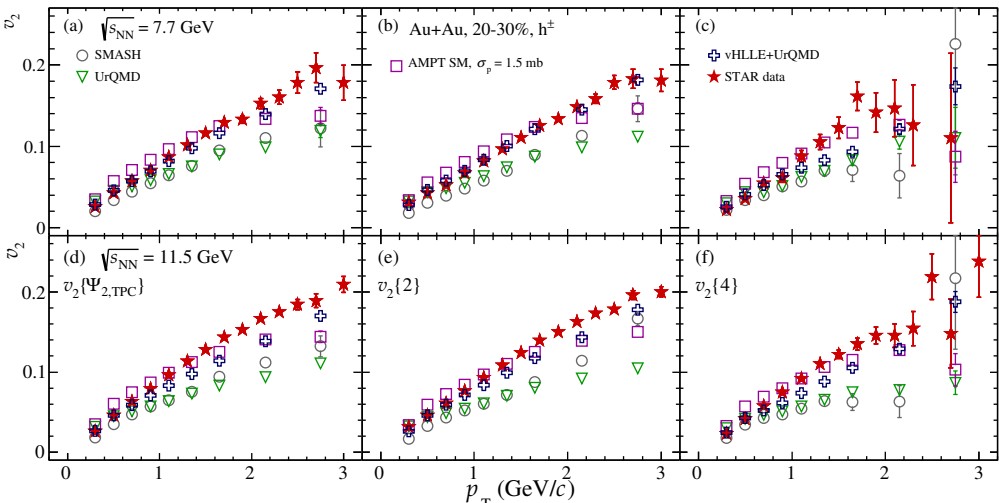

**Figure 2.** $p_T$-dependence of $v_2$ of inclusive charged hadrons from 20–30% centrality Au+Au collisions at $\sqrt{s_{NN}} = 7.7$ GeV (upper row) and 11.5 GeV (lower row) obtained using $v_2\{\Psi_{2,\text{TPC}}\}$ (**a,d**), $v_2\{2\}$ (**b,e**), and $v_2\{4\}$ (**c,f**) methods of flow measurements in comparison with STAR data [13].

Figure 3a demonstrates the centrality dependence of the $v_2\{4\}/v_2\{2\}$ ratio in Au+Au collisions at $\sqrt{s_{NN}} = 11.5$–39 GeV. Data points of $v_2\{4\}$ and $v_2\{2\}$ are taken from Ref. [13]. The $v_2\{4\}/v_2\{2\}$ ratio weakly depends on the colliding energy. It also exposes a specific centrality dependence: $v_2\{4\}/v_2\{2\}$ increases from most central to mid-central collisions.

Recall that a small $v_2\{4\}/v_2\{2\}$ ratio corresponds to large $v_2$ fluctuations. This centrality dependence is consistent with the pattern, where the initial eccentricity $\varepsilon_2$ fluctuations and $v_2$ fluctuations, in their turn, are expected to dominate in most central collisions and decrease their contribution towards mid-central collisions.

Figure 3b–f shows the $v_2\{4\}/v_2\{2\}$ ratio obtained from the models at $\sqrt{s_{NN}} = 7.7$–39 GeV. Although the UrQMD and SMASH models failed to reproduce the observed $v_2$ as shown in Figure 2, they predict similar values of the $v_2\{4\}/v_2\{2\}$ ratio to those from vHLLE+UrQMD, AMPT SM, and STAR data. These results again support that the $v_2$ fluctuations are mainly driven by the initial eccentricity $\varepsilon_2$ fluctuations.

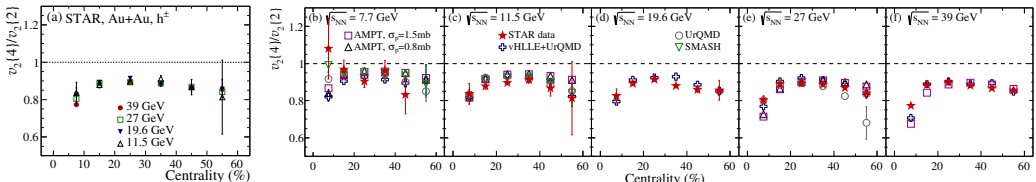

**Figure 3.** Centrality dependence of the $v_2\{4\}/v_2\{2\}$ ratio of charged hadrons from Au+Au collisions at $\sqrt{s_{NN}} = 39$–11.5 GeV (**a**) measured in STAR [13], and its comparison with vHLLE+UrQMD, AMPT SM, UrQMD and SMASH models at $\sqrt{s_{NN}} = 7.7$–39 GeV (**b**–**f**).

Figure 4 shows the centrality dependence of $v_2$ in Au+Au collisions from the SMASH, UrQMD, AMPT SM, and vHLLE+UrQMD models at $\sqrt{s_{NN}} = 7.7$ and 11.5 GeV. On the one hand, all of the two-particle correlation based methods ($v_2\{\Psi_{2,\mathrm{TPC}}\}$, $v_2^{\mathrm{SP}}\{Q_{2,\mathrm{TPC}}\}$, $v_2\{2\}$) are in excellent agreement across centrality classes. On the other hand, the multi-particle correlation based methods ($v_2\{4\}$, $v_2\{\mathrm{LYZ}\}$) and $v_2$ measured with respect to the event plane of the first order harmonic ($v_2^{\mathrm{SP}}\{Q_{1,\mathrm{FHCal}}\}$, $v_2\{\Psi_{1,\mathrm{FHCal}}\}$) agree with each other and their ratios to $v_2\{2\}$ have shown the peculiar centrality dependence observed in Figure 3.

Due to small $v_2$ magnitude and large fluctuations, the four-particle $Q$-cumulant fails in the 0–5% most central collisions. Hence, these data points are excluded from the ratio panels. For this reason, $v_2^{\mathrm{SP}}\{Q_{1,\mathrm{FHCal}}\}$ and $v_2\{\Psi_{1,\mathrm{FHCal}}\}$ may serve as important alternatives to estimate the magnitude of the $v_2$ fluctuations in the most central collisions. Unfortunately, for the Lee–Yang zeros method, due to low resolution parameter $\chi$ as mentioned in the previous section, the $v_2\{\mathrm{LYZ}\}$ only succeeded in measuring $v_2$ in 10–40% mid-central Au+Au collisions for the AMPT SM model.

Figure 5 shows the $p_T$-dependence of charged hadron $v_2$ in 10–40% mid-central Au+Au collisions at $\sqrt{s_{NN}} = 7.7$ and 11.5 GeV. The AMPT SM and vHLLE+UrQMD models predict a weak $p_T$-dependence of the $v_2\{4\}/v_2\{2\}$ cumulant ratio, which is again consistent with the dominant initial eccentricity $\varepsilon_2$ fluctuation pattern. On the other hand, the $v_2\{\mathrm{LYZ}\}$ shows a good agreement with $v_2\{4\}$ for the AMPT model. Lee–Yang zeros method appears fairly feasible for $v_2$ measurements only in mid-central collisions at the NICA energy range.

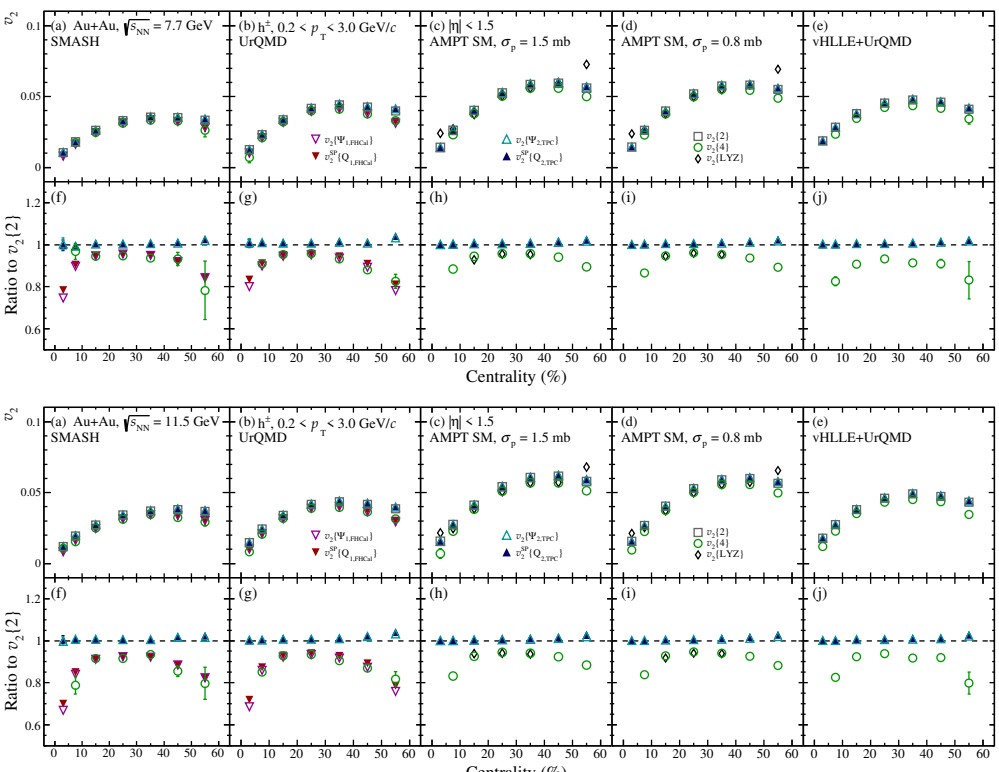

**Figure 4.** Centrality dependence of $v_2$ of inclusive charged hadrons from Au+Au collisions at $\sqrt{s_{NN}}$ = 7.7 and 11.5 GeV measured by different methods. Panels (**a**–**e**) correspond to different models. Panels (**f**–**j**) show the $v_2\{\text{method}\}/v_2\{2\}$ ratio.

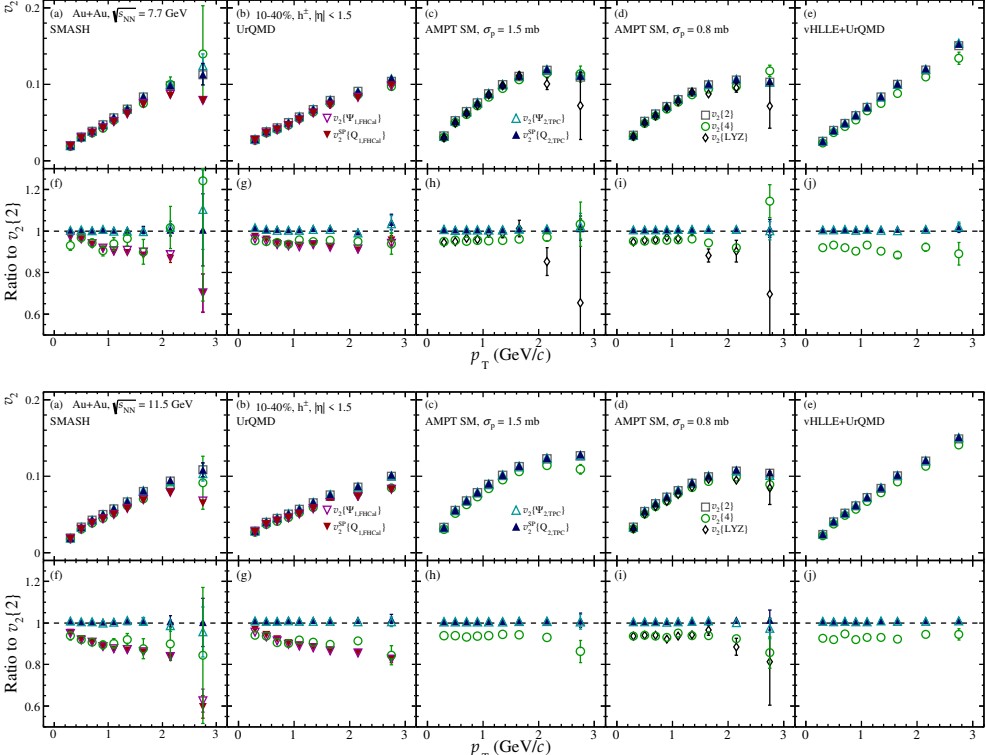

**Figure 5.** $p_T$-dependence of $v_2$ of inclusive charged hadrons in 10–40% central Au+Au collisions at $\sqrt{s_{NN}}$ = 7.7 and 11.5 GeV measured by different methods. Panels (**a**–**e**) correspond to different models. Panels (**f**–**j**) show the $v_2\{\text{method}\}/v_2\{2\}$ ratio.

Figure 6 shows the centrality and $p_T$-dependence of the $v_2\{4\}/v_2\{2\}$ ratio for different particle species: pions, kaons and protons. The SMASH model is excluded from this figure due to its limited statistics. The weak dependence of the $v_2\{4\}/v_2\{2\}$ ratio on $p_T$ is again observed. Furthermore, we found the $v_2\{4\}/v_2\{2\}$ ratio to weakly depend on particle species.

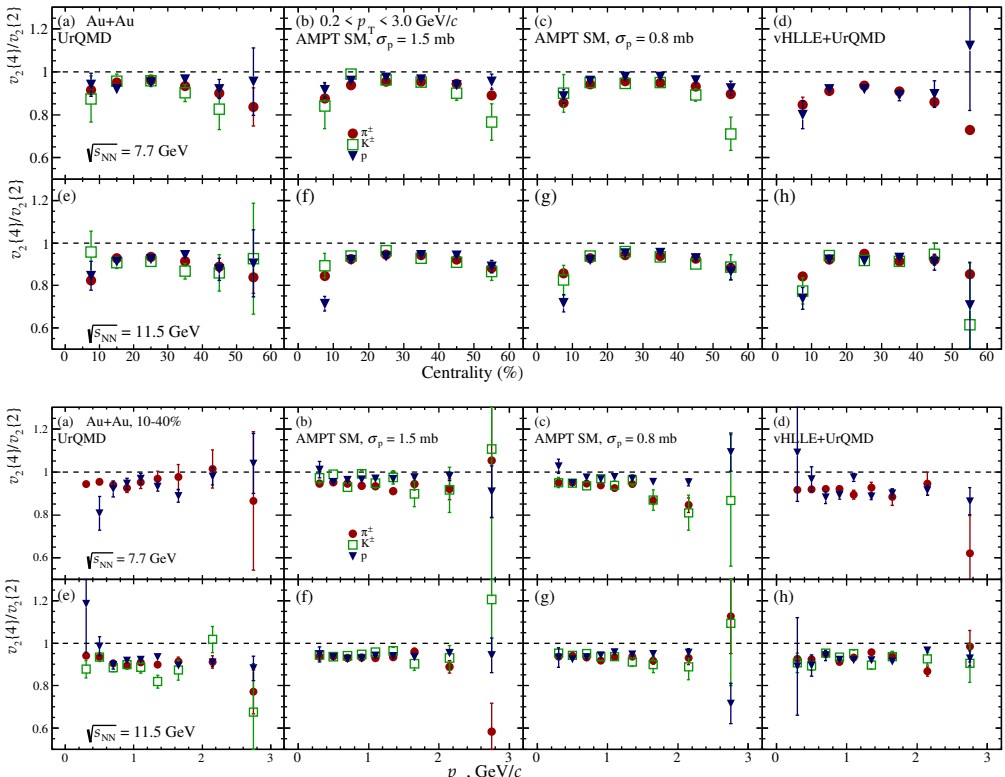

**Figure 6.** Centrality and $p_T$ dependences of relative elliptic flow fluctuations $v_2\{4\}/v_2\{2\}$ of identified hadrons from Au+Au collisions at $\sqrt{s_{NN}}$ = 7.7 GeV (**a–d**) and 11.5 GeV (**e–h**). Columns correspond to different used models.

Figure 7 shows the comparison between high-order $Q$-cumulants: $v_2\{6\}$ and $v_2\{4\}$. $v_2\{4\}$ agrees with $v_2\{6\}$ within statistical errors. It should be noted that, for their measurements, especially with the NICA energy regime, large statistics is required. The observed fine splitting between $v_2\{6\}$ and $v_2\{4\}$ at the percent level can be interpreted as a signature of the deviation of $v_2$ p.d.f. from the Gaussian form [31].

Figure 8 shows the prospects of the $v_2$ measurements in the future MPD experiment at the NICA collider. The open markers depict $v_2$ measurements from the UrQMD event generator, while the filled markers correspond to the measurements from the data that have undergone the full realistic reconstruction chain with MPD detector geometry simulated in GEANT4. The good agreement between the two sets of measurements for both pions and protons allows one to be assured of the precise future flow measurements in the MPD experiment at NICA.

Figure 9 shows that the methods for flow measurements are robust towards non-uniform acceptance. We have simulated the non-uniformity, which may occur during data taking: the malfunction of TPC sectors and FHCal modules. The effect of non-uniform acceptance is expected to be smaller than 2% for all the flow measurement methods.

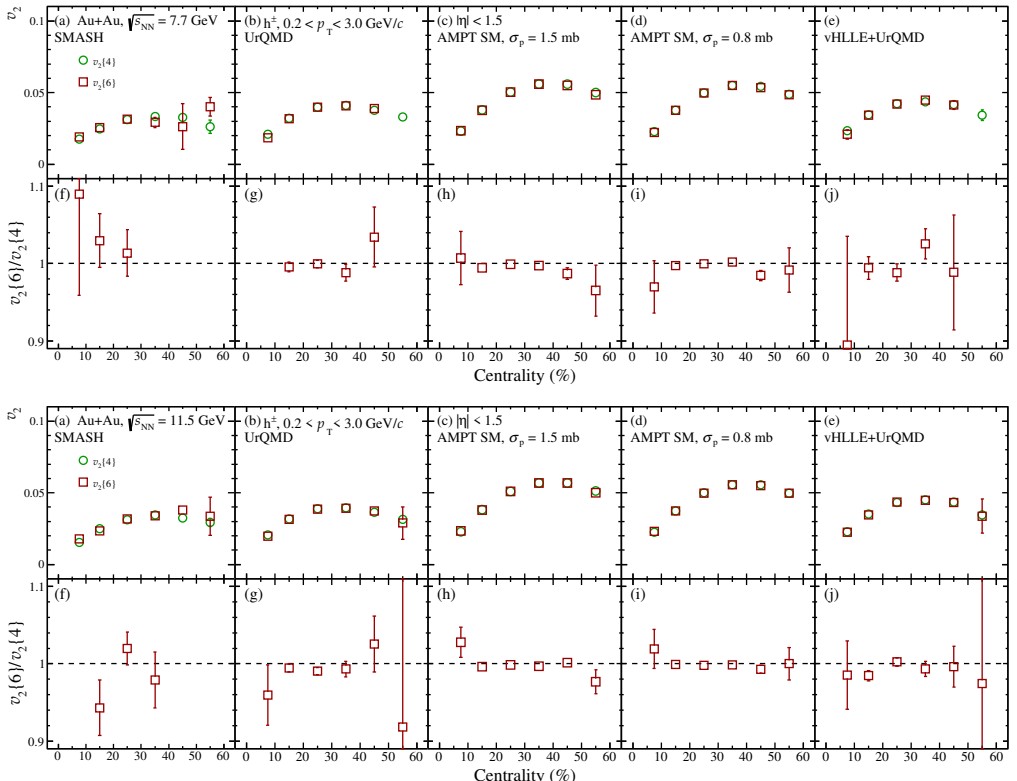

**Figure 7.** Centrality dependence of $v_2$ of inclusive charged hadrons from Au+Au collisions at $\sqrt{s_{NN}} = 7.7$ and 11.5 GeV obtained using the high-order Q-cumulants $v_2\{4\}$, $v_2\{6\}$ methods in different models (**a**–**e**). Panels (**f**–**j**) show the ratio $v_2\{6\}/v_2\{4\}$.

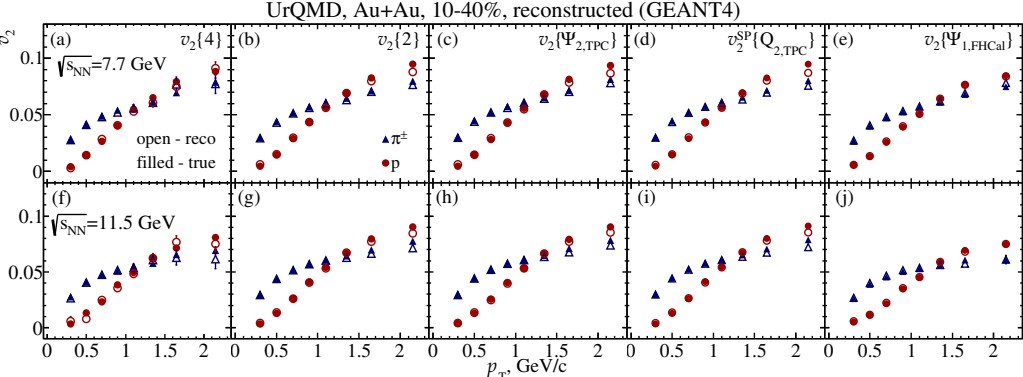

**Figure 8.** Comparison of $v_2(p_T)$ for pions and protons in 10–40% mid-central Au+Au collisions at $\sqrt{s_{NN}} = 7.7$ GeV and $\sqrt{s_{NN}} = 11.5$ GeV obtained by four-particle cumulants (**a**,**f**), two-particle cumulants (**b**,**g**), TPC event plane (**c**,**h**), TPC scalar product (**d**,**i**), FHCal event plane (**e**,**j**) methods of fully reconstructed ("reco") and generated UrQMD events ("true").

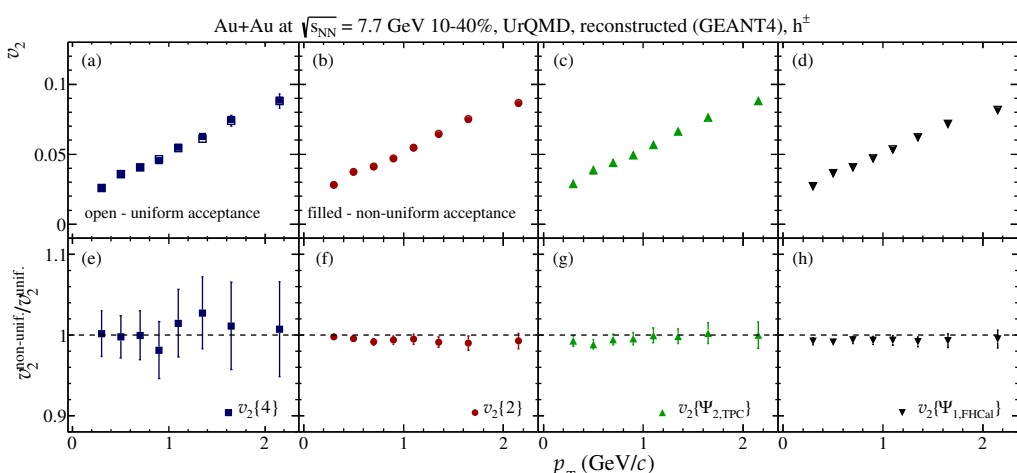

**Figure 9.** Comparison of $v_2(p_T)$ of cases with uniform acceptance (open markers) and with a "hole" in the TPC azimuthal acceptance (closed markers) for charged hadrons in 10–40% mid-central Au+Au collisions at $\sqrt{s_{NN}} = 7.7$ GeV obtained by four-particle $Q$-cumulants (**a**), two-particle $Q$-cumulants (**b**), TPC (**c**) and FHCal (**d**) event planes. Lower panels (**e–h**) show the ratio of the non-uniform case to the uniform one.

## 5. Conclusions

In this work, we have shown that the $v_2$ fluctuations measured via the $v_2\{4\}/v_2\{2\}$ cumulant ratio suggest that their main source is the eccentricity driven fluctuations in the initial-state geometry. At $\sqrt{s_{NN}} = 7.7$ and 11.5 GeV, the $v_2$ measurements with respect to the participant plane can shed light on the $v_2$ fluctuations in centrality ranges, where the $v_2\{4\}$ is not applicable. In the future, we plan to include the data from other transport models and extend the study to other colliding energies and systems.

The performance of the MPD experiment's elliptic flow measurements has been studied with Monte-Carlo simulations using Au+Au at NICA energies. A large sample of generated UrQMD minimum bias events was used as an input for the full chain of realistic simulations of the MPD detector subsystems based on the GEANT4 platform and reconstruction algorithms built in the MPDROOT. Realistic procedures for centrality determination, particle identification and event plane reconstruction were applied in the analysis. The resulting performance of the MPD was verified for $v_2$ measurements of identified charged pions, kaons, and protons as a function of transverse momentum in different centrality classes. The detailed comparison of the results obtained from the analysis of the fully reconstructed data and generator-level data has allowed us to conclude that the MPD system will provide detailed differential measurements of elliptic flows with high efficiency.

**Author Contributions:** All the authors contributed equally to this work. All authors have read and agreed to the published version of the manuscript.

**Funding:** This research was supported by MEPhI program Priority 2030 and by the Ministry of Science and Higher Education of the Russian Federation, Project "New Phenomena in Particle Physics and the Early Universe" No FSWU-2023-0073.

**Institutional Review Board Statement:** Not applicable.

**Informed Consent Statement:** Not applicable.

**Data Availability Statement:** The data presented in this study are available on request from the corresponding author. The data are not publicly available since they contains the data produced by the internal software of the experiment.

**Acknowledgments:** The authors thank Andrey Moshkin from the MPD collaboration for help with the production of the reconstructed data used in the analysis. Computational resources were provided

by the NRNU MEPhI high-performance computing center and NICA high-performance cluster of LHEP JINR.

**Conflicts of Interest:** The authors declare no conflict of interest. The funders had no role in the design of the study; in the collection, analyses, or interpretation of data; in the writing of the manuscript, or in the decision to publish the results.

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
