# Peer review of "Elliptic Flow and Its Fluctuations from Transport Models for Au+Au Collisions at sNN = 7.7 and 11.5 GeV"

_2571-712X, doi:10.3390/particles6010002_

Round 1

Reviewer 1 Report

I find that this manuscript is a) timely b) interesting and c) important enough for a publication in the MDPI Journal Particles. The scientific content is timely as the MPD detector at the NICA accelerator at JINR, Dubna is scheduled to be completed in 2023. The manuscript reads well, I have detected only a few typos that are to be corrected before publication. These are collected at the end of my report. 

The introduction reads well and it indicates the timeliness of the reported research, which is a preparation study for the operation of the MPD detector at the upcoming NICA accelerator at JINR, Dubna. The second chapter of this manuscript describes several event generators (UrQMD, SMASH, APMT+SM and vHLLE+UrQMD) and summarizes well their key physics ingredients and their relations to other Monte-Carlos and components (eg. HIJING, ZPC, ART etc). The third section deals with summarizing extensively the different measurement methods for elliptic flow measurements. Section 4 summarizes the results and discussions. 

The main conclusions are shown in Figs. 8 and 9. Fig. 8 indicates, that the detailed detector level simulations with UrQMD + Geant4 agree reasonably well with the UrQMD Monte-Carlo level simulations and assure the possibility of future precise flow measurements with the MPD detector at NICA both for pions and for protons at two different NICA energies and using five different methods for flow determination.

Fig. 9 indicates, that possible detector acceptance limitations will not influence critically the precision of the foreseen measurements, provided that these acceptance limitations are sufficiently small.

These conclusions are timely and important, hence it is my pleasure to recommend this manuscript for an acceptance and a publication in the MDPI Journal Particles.

However, before publication, I ask the authors to fix the following minor typos in the current text of the manuscript:

Line 18:

v2 signal --> The elliptic flow v2 signal

(do not start a sentence with a formula, but with text)

Line 21:

after sQGP, add a reference for this discovery (e.g. the White Papers of the four RHIC collaborations, or the paper of Shuryak that proposed this interpretation)

line 127:

sufficient --> sufficiently large

line 140:

Start the new paragraph with a tabulator (seems to be a small formatting error).

Line 182:

Please check the start of this sentence, a possible, suggested fix is as follows:

Following the above-mentioned descriptions, ...

Figures:

Make sure to add sufficient amount of white space after each of the figure captions, eg. Figs. 2, 3, 6.

line 258

has been --> have been

After these minor corrections, I recommend publishing this manuscript in the MDPI Journal Particles. I trust that the Authors will implement the suggested minor corrections faithfully. I do not have to review this manuscript again.

Author Response

Thank you very much for your remarks.

We have corrected the grammar mistakes and added the reference to the paper of Shuryak.

Reviewer 2 Report

In this manuscript the author use a number of transport (and hydrodynamic) models to compute various measures of elliptic flow in Au+Au collisions at 7.7 and 11.5 GeV per nucleon pair, comparing to measurements performed at RHIC by the STAR Collaboration.   The main goal is to study the feasibility and performance of potential measurements that could be made by the MPD at the NICA Collider.  They find that elliptic flow measurements should indeed be feasible.  Further, they verify that in these models (which are compatible with STAR data), fluctuations of elliptic flow largely follow the initial geometric eccentricity, even at these collision energies.

The methods are sounds, and the results look reasonable.  While there is nothing groundbreaking, the results should be useful for future measurements at colliders such as NICA.    I recommend publication after the authors consider the following comments.

In the abstract, "provide the direct comparison with published results" should read "provide a direct comparison with published results".

In the in-line equation from the second sentence of the introduction, elliptic flow (v2) is defined, restricting to the second harmonic since that is the all that is studied in this work.  However, the right side of the equation uses \Psi_n, referencing an arbitrary harmonic n.  Presumably this should read \Psi_2 (and similarly later in the sentence).

However, regarding this point, it would be useful to include a comment about why only v_2 is studied.   While Ref. [12] only presents elliptic flow data, there has subsequently been much study and various measurements of other harmonics, including at these energies.  (See, for example, arXiv:1909.09640).  Presumably it would be interesting to know how the feasibility of these other observables in future colliders.   While it may not be feasible to include every possible observable in this work (rapidity-odd directed flow, mixed-harmonic correlations, etc.), presumably it would not be difficult to replace, e.g., n=2 with n=3, etc., in the observables already studied here.  Presumably, the statistical requirements and performance would be different.   Was it found to be unfeasible, or did the authors not think it useful/interesting?  In either case, it would be useful to include a statement in the text.

Author Response

Thank you for your remarks.

Please find the reply on your comments below:

  1. We have corrected the grammar mistake.
  2. In this work, we have shown the study of elliptic flow v_2 with respect not only to the second order event plane \Psi_2, but also to the first order event plane \Psi_1. Following this, the \Psi_n notation has been chosen.
  3. Because the focus of this work is the elliptic flow fluctuations, we did not include other flow harmonics to this study. While there are studies on the fluctuations of the triangular flow v_3 at LHC energies (see Ref. [30] for example), it is known that the multi-particle cumulants method used for the flow fluctuation study reaches its limit for the low multiplicity at NICA energies and smaller v_3 than v_2 signal.

Reviewer 3 Report

1. The authors could emphasize the main physical focus of NICA facility while conducting the elliptic flow and fluctuation. 

2. The authors can add some discussions on the possible effects of hadronic EoS as well as the momentum-dependence of the single baryon potential on the relevant observables.

3. Also the possible phase transition would affect the results of  the models used in the present study should be discussed.

After considering the above potential effects, the manuscript could be accepted.

Author Response

Thank you for your remarks.

Please find the reply on your comments below:

  1. We have added the following sentence to the “Introduction” section: “The primary scientific mission of the MPD experiment is to investigate the properties of strongly interacting matter at high net-baryon densities.”  
  2. In the present work we do not study the possible effects of hadronic EoS as well as the momentum-dependence of the single baryon potential on the relevant observables, but it can be a good topic for the futher work.

  3. Our study shows that the main source of the elliptic flow fluctuations at  \sqrt{s_NN} = 7-11 GeV is the fluctuations of the nucleons-participants in the initial stages of relativistic heavy-ion collisions. Thus, given experimental observable may not be sensitive to the phase transition.